# Clinical Differentiation of Severe Fever with Thrombocytopenia Syndrome from Japanese Spotted Fever

**DOI:** 10.3390/v14081807

**Published:** 2022-08-18

**Authors:** Nana Nakada, Kazuko Yamamoto, Moe Tanaka, Hiroki Ashizawa, Masataka Yoshida, Asuka Umemura, Yuichi Fukuda, Shungo Katoh, Makoto Sumiyoshi, Satoshi Mihara, Tsutomu Kobayashi, Yuya Ito, Nobuyuki Ashizawa, Kazuaki Takeda, Shotaro Ide, Naoki Iwanaga, Takahiro Takazono, Masato Tashiro, Takeshi Tanaka, Seiko Nakamichi, Konosuke Morimoto, Koya Ariyoshi, Kouichi Morita, Shintaro Kurihara, Katsunori Yanagihara, Akitsugu Furumoto, Koichi Izumikawa, Hiroshi Mukae

**Affiliations:** 1Department of Respiratory Medicine, Nagasaki University Hospital, Nagasaki 852-8102, Japan; 2Health Center, Nagasaki University, Nagasaki 852-8521, Japan; 3Department of Respiratory Medicine, Sasebo City General Hospital, Sasebo 857-8511, Japan; 4Department of General Internal Medicine, Nagasaki Rosai Hospital, Sasebo 857-0134, Japan; 5Department of Respiratory Medicine, Isahaya General Hospital, Isahaya 854-8501, Japan; 6Department of Respiratory Medicine, Sasebo Chuo Hospital, Sasebo 857-1195, Japan; 7Department of Infection Control and Education Center, Nagasaki University Hospital, Nagasaki 852-8102, Japan; 8Infectious Disease Experts Training Center, Nagasaki University Hospital, Nagasaki 852-8102, Japan; 9Department of Clinical Medicine, Institute of Tropical Medicine, Nagasaki University, Nagasaki 852-8523, Japan; 10Department of Virology, Institute of Tropical Medicine, Nagasaki University, Nagasaki 852-8523, Japan; 11Department of Medical Safety, Nagasaki University Hospital, Nagasaki 852-8102, Japan; 12Department of Laboratory Medicine, Nagasaki University Hospital, Nagasaki 852-8102, Japan

**Keywords:** severe fever with thrombocytopenia syndrome, Japanese spotted fever, clinical differentiation, white blood cell

## Abstract

Severe fever with thrombocytopenia syndrome (SFTS) and Japanese spotted fever (JSF; a spotted fever group rickettsiosis) are tick-borne zoonoses that are becoming a significant public health threat in Japan and East Asia. Strategies for treatment and infection control differ between the two; therefore, initial differential diagnosis is important. We aimed to compare the clinical characteristics of SFTS and JSF based on symptomology, physical examination, laboratory data, and radiography findings at admission. This retrospective study included patients with SFTS and JSF treated at five hospitals in Nagasaki Prefecture, western Japan, between 2013 and 2020. Data from 23 patients with SFTS and 38 patients with JSF were examined for differentiating factors and were divided by 7:3 into a training cohort and a validation cohort. Decision tree analysis revealed leukopenia (white blood cell [WBC] < 4000/μL) and altered mental status as the best differentiating factors (AUC 1.000) with 100% sensitivity and 100% specificity. Using only physical examination factors, absence of skin rash and altered mental status resulted in the best differentiating factors with AUC 0.871, 71.4% sensitivity, and 90.0% specificity. When treating patients with suspected tick-borne infection, WBC < 4000/µL, absence of skin rash, and altered mental status are very useful to differentiate SFTS from JSF.

## 1. Introduction

Severe fever with thrombocytopenia syndrome (SFTS) is a tick-borne infectious disease caused by the SFTS virus (SFTSV or *Dabie bandavirus*, a novel genus *Banyangvirus* of the family *Phenuiviridae* that was first reported in China in 2011 [1,2]. Since then, many cases have been reported in west Asia, China, Korea, and Japan [3] and are classified as viral hemorrhagic fever [4]. The mortality rate of SFTS ranges from 5–40% [4,5,6,7,8]. Around 60–100 cases of SFTS have been reported annually since 2013 in Japan [9], with endemic areas localized in western Japan.

Spotted fever group rickettsioses are another emerging tick-borne infectious disease that are widely distributed throughout the world, including Rocky Mountain spotted fever in North America, Mediterranean spotted fever on the Mediterranean coast, Queensland tick typhus in Australia, and Japanese spotted fever (JSF) in Japan [10]. JSF is caused by Alphaproteobacteria, belonging to the spotted fever group of the genus Rickettsia [10]. JSF was first reported in Tokushima Prefecture, western Japan, in 1984 [11]. Since then, it has been reported in South Korea, Thailand, the Philippines, and China [12,13], and its related species have also been reported [14,15]. The number of JSF cases in Japan has increased since 2016 to 250–350 cases reported annually [16,17] with a mortality rate from 1–4% [17,18].

Although scrub typhus, another tick-borne infection caused by *Orientia tsutsugamushi,* is reported in 400–500 cases annually across Japan throughout the year [19], SFTS and JSF are major tick-borne infections especially prevalent in western Japan during the summer season. They are category IV infectious diseases that require the reporting of all cases under the Japanese Infectious Diseases Control Law (criteria for notification: [20,21]). The vector tick species that carry both SFTS and JSF are *Haemaphysalis longicornis*, *H. flava*, and *Amblyomma testudinarium* [1,17,22,23,24], and the two tick-borne infections have similar epidemiological and geographic distributions [16]. Despite similar clinical manifestations such as fever, tick bites, and neurological and gastrointestinal symptoms [5,17,25,26], different treatment regimens and infection-control strategies are applied to each infection; therefore, clinical differentiation between SFTS and JSF is particularly important [4,17]. JSF may be treated with antibiotics; however, effective antiviral therapy has not been established for SFTS. Human-to-human transmission of SFTSV may occur through blood or bodily secretions [27,28,29,30]. Additionally, their diagnosis requires serological or polymerase chain reaction (PCR) testing to confirm the causative pathogens, which are not easily accessed and require time to perform and analyze [8]. Early diagnosis of SFTS and JSF is important due to the potentially fatal nature of the diseases. Differential diagnosis at the initial clinical presentation is useful in planning the management of treatment and infection control.

In this study, we conducted a retrospective comparison of clinical manifestations, laboratory data, and radiologic features of patients with SFTS and JSF hospitalized in Nagasaki, a southwestern prefecture of Japan. Based on decision tree analysis, we identified potential significant factors for differential diagnosis of SFTS and JSF.

## 2. Materials and Methods

### 2.1. Study Population and Clinical Data

We retrospectively reviewed the medical records of patients aged ≥20 years diagnosed with laboratory-confirmed SFTS and JSF between January 2013 and December 2020 at five hospitals (two tertiary medical institutions: Nagasaki University Hospital and Sasebo City General Hospital; three secondary medical institutions: Isahaya General Hospital, Sasebo Chuo Hospital, and Nagasaki Rosai Hospital) in Nagasaki Prefecture, Japan. The medical records of all enrolled patients were reviewed. Data on baseline characteristics, clinical presentations, laboratory findings, and radiological findings at admission were collected from the medical records. In-hospital complications and outcomes were assessed.

### 2.2. Definitions

All definitions used in the study were established before the data analyses. An episode of SFTS was defined as SFTSV, in which viral RNA was confirmed by real-time PCR analysis of a serum sample, as previously described [31]. The diagnosis of JSF was defined as either a 4-fold increase in immunoglobulin G (IgG) in the serum sample with immunofluorescence assay, PCR analysis of a blood sample, or tissue biopsy (eschar) [32]. Altered mental status was defined as Japan coma scale ≥1 points [33]. Acute kidney injury (AKI) was diagnosed according to the Kidney Disease: Improving Global Outcomes definition, as described previously [34]. Chronic kidney disease (CKD) was defined as a reduced glomerular filtration rate (<60 mL/min/1.73 m^2^) for more than 3 months. [35] Immunosuppressants included corticosteroids and other immunosuppressive agents. Disseminated intravascular coagulation (DIC) scores were calculated using the Japanese Association for Acute Medicine (JAAM) DIC scoring system [36]. Hemophagocytic lymphohistiocytosis (HLH) was defined using HLH-2004 criteria [37].

Complications and treatments were respectively determined as those that developed or were initiated after the initial visit or the date of hospital admission.

### 2.3. Evaluation of the Chest Radiograph and Computed Tomography (CT)

Two Japanese Respiratory Society (JRS) board-certified pulmonologists each with 15 years of experience reviewed all chest radiographs and CT examinations independently and arrived at a consensus. Cardiomegaly was defined as a cardiothoracic (CTR) ratio > 0.50, based on posterior-anterior (PA) chest radiographs, or a CTR ratio > 0.55 on anterior-posterior (AP) chest radiographs [31]. Abnormalities on chest CT were characterized by consolidation, ground-glass opacity (GGO), centrilobular nodules, interstitial septal thickening, and bronchial wall thickening. The presence of mediastinal lymph node enlargement (>10 mm along the short axis), pleural effusion, pericardial effusion (pericardial thickness of 4 mm or more [31]), hepatomegaly (diameter of >16.0 cm at craniocaudal line [31]), splenomegaly (width measurement of >10.5 cm [31]), and additional lung findings were also recorded.

### 2.4. Statistical Analysis

The results are expressed as means ± standard deviations and as percentages. Categorical variables were analyzed using Fisher’s exact test, and continuous variables were analyzed using a t-test or Wilcoxon rank sum test. All tests were two-tailed, and differences were considered significant at a *p*-value < 0.05 or less than the value adjusted by Bonferroni correction.

All cases were divided by 7:3 into a training cohort and a validation cohort. In the training data, a decision tree was built to differentiate SFTS from JSF using variables that were significantly different on bivariate analysis and missing values less than 20%. In addition, we performed a decision tree on each continuous variable one by one to determine cut off values and applied them to categorical data.

The discriminatory power of the differentiation models was evaluated by assessing the area under the receiver operating characteristic curve (AUC), sensitivity, specificity, positive predictive value, and negative predictive value. All analyses were performed using JMP^®^ Pro 16.2.0 predictive analysis software (SAS Institute Inc., Cary, NC, USA).

### 2.5. Ethical Considerations

The study protocol was approved by the Institutional Review Board of Nagasaki University Hospital (approval number 18121024), Isahaya General Hospital (approval number, 2020-21), Sasebo City General Hospital (approval number 2020-A024), Sasebo Chuo Hospital (approval number 2020-17), and Nagasaki Rosai Hospital (approval number 02003). The institutional review board waived the requirement for informed consent from patients included in this study because of its retrospective design. The summary and information of this study have been disclosed, following the regulations of each participating hospital.

## 3. Results

### 3.1. Study Population

Data from a total of 61 hospitalized patients (23 with SFTS and 38 with JSF) at five hospitals during the study period were reviewed. All 23 cases of SFTS were confirmed by PCR testing of blood samples; 2 cases of JSF were confirmed by both the presence of increased *Rickettsia japonica* antibody levels and PCR test of blood and/or tissue sample, 12 cases by antibody test only, and the remaining 24 cases by PCR testing of blood and/or tissue samples. One case of JSF, diagnosed by antibodies to *R. japonica* and PCR of blood, was a case of internationally imported infection, with elevated antibodies to *R. cornorii* and *R. africae* other than *R. japonica*, and the coexistence of rickettsiosis infection was suspected.

### 3.2. Patients’ Baseline Characteristics

The baseline patient characteristics are presented in Table 1. The elderly population was predominant in both the SFTS (mean 70.6 years) and JSF groups (mean 68.8 years). There were no significant differences between the two groups in terms of male sex or underlying diseases. Immunosuppressants were used in two cases of SFTS: corticosteroids and adalimumab. Patients who were farmers, hunters, or living or working in wooded and hilly areas comprised 50.0% of SFTS patients and 27.8% of JSF patients. Patients in both groups were diagnosed frequently in spring and summer, though incidence of SFTS was lower (8.7%) in autumn compared to that of JSF (34.2%). No patients were diagnosed during winter.

### 3.3. Patients’ Clinical Symptoms

Table 2 presents the clinical symptoms and physical findings of patients with SFTS or JSF at admission. Approximately 5 and 6 days in patients with SFTS and JSF, respectively, passed between onset of illness and admission. Patients with SFTS were significantly more frequently reporting altered mental status (60.9% vs. 21.1%, *p* = 0.003) and diarrhea (52.4% vs. 10.5%, *p* = 0.001) compared to patients with JSF. In contrast, skin rash (34.8% vs. 100%, *p* < 0.0001) and tick bites (42.9% vs. 88.9%, *p* = 0.001) were more frequently observed in patients with JSF. With regard to vital signs, most patients in both groups were febrile. There was no difference between the two groups in terms of respiratory rate or SpO_2_/FiO_2_ ratio. The qSOFA scores were comparable between the two groups.

### 3.4. Labortory Findings

The laboratory findings on admission are presented in Table 3. With regard to peripheral blood cell counts, patients with SFTS showed significantly lower numbers of white blood cells (WBCs; mean: 1821.3/μL vs. 7383.9/μL, *p* < 0.0001) and neutrophils (mean; 1190.8/μL vs. 6176.0/μL, *p* < 0.0001) compared to those in patients with JSF. Higher atypical lymphocyte ratio (mean: 2.0% vs. 0.2%, *p* = 0.001) and thrombocytopenia (mean: 58.4 × 10^3^/μL vs. 113.1 × 10^3^/μL, *p* = 0.001) were observed in the SFTS group. Inflammatory markers such as C-reactive protein (CRP; mean: 0.8 mg/dL vs. 12.3 mg/dL, *p* < 0.0001), and soluble IL-2 receptor (sIL-2R; mean: 1451.6 IU/L vs. 4521.0 IU/L, *p* = 0.017) were higher in the JSF group. Highly elevated liver enzymes including aspartate aminotransferase (AST; mean: 328.3 IU/L vs. 88.9 IU/L, *p* < 0.0001), alanine aminotransferase (ALT; mean: 113.5 IU/L vs. 54.2 IU/L, *p* = 0.001), and lactate dehydrogenase (LDH; mean: 801.5 IU/L vs. 389.4 IU/L, *p* = 0.001) were more prevalent in the SFTS group compared to those in the JSF group. Prolonged activated partial thromboplastin time (APTT; mean: 48.0 s vs. 36.9 s, *p* = 0.001) was seen in the SFTS group compared to the JSF group, though fibrinogen was higher in JFS group (mean: 212.7 mg/dL vs. 382.6 mg/dL, *p* = 0.001).

### 3.5. Radiologic Findings

The radiological findings are presented in Table 4. Chest radiograph was performed in 58 patients (95.1%). Chest CT examinations were performed in 22 patients (95.7%) with SFTS and 17 patients (44.7%) with JSF. CT scans were performed at an average of 5–6 days after symptom onset, with an average of 0–1 day after admission with 1 mm (*n* = 28), 2 mm (*n* = 5), 3 mm (*n* = 2), or 5 mm (*n* = 4) slices at the spine position.

A total of 19 patients with SFTS (86.4%) and 13 patients (76.5%) with JSF had abnormal chest CT findings. Interstitial septal thickening (68.2% vs. 64.7%) and ground glass opacity (GGO) (54.6% vs. 52.9%) were the most frequently found chest CT abnormalities in SFTS and JSF patients, though no differences between groups were identified. Findings of centrilobular nodules and bronchial wall thickening were likely more frequent in the SFTS group (36.4% vs. 11.8%, *p* = 0.140, 36.4% vs. 5.9%, *p* = 0.052) than the JSF group. Pleural effusion was likely more frequently detected in patients with JSF than those with SFTS (22.7% vs. 58.8%, *p* = 0.045). Enlarged mediastinal lymph nodes were detected only in patients with SFTS (13.6%).

### 3.6. Outcomes and Treatments

Appendix A shows the outcomes, complications, and treatments of the patients with SFTS and JSF. The mortality rate of patients with SFTS (26.1%) was higher that of patients with JSF (0.0%) (*p* = 0.002). The length of hospital stay was longer (mean; 34.5 days vs. 15.9 days, *p* = 0.015) and the ICU admission rate was higher (mean; 30.4% vs. 7.9%, *p* = 0.032) in patients with SFTS than those in patients with JSF.

With regard to in-hospital complications, secondary bacterial infections occurring more frequently in the SFTS group than in the JSF group (mean: 26.1% vs. 5.3%, *p* = 0.044) consisted of bacteremia (*n* = 4) caused by *Streptococcus epidermidis* or methicillin-resistant *S. capitis,* followed by pneumonia (*n* = 2), and urinary tract infections (*n* = 1) caused by *Pseudomonas aeruginosa*. Secondary fungal infections, although not statistically significant, were also more frequent in patients with SFTS (17.4%). These infections include invasive aspergillosis, disseminated trichosporonosis, candidiasis, and pulmonary cryptococcosis. One case of SFTSV meningitis and another case of rickettsial meningitis were observed in the study population. A total of 11 (47.8%) and 16 (69.6%) cases of patients with SFTS developed hemophagocytic lymphohistiocytosis (HLH) or disseminated intravascular coagulation (DIC), respectively. The diagnosis of 9 of 11 cases of HLH in patients with SFTS was made by bone marrow biopsy.

All patients with SFTS were administered antibiotics against rickettsial infections at admission until the SFTSV diagnosis was confirmed. Antiviral agents such as favipiravir and ribavirin were used in five (27.3%) and two (8.7%) patients with SFTS, respectively, based on clinical trials or off-label use [26,38]. All patients with JSF received tetracyclines, including minocycline in 36 cases, doxycycline in 1 case, and a combination of minocycline and doxycycline in 1 case. A total of 12 (31.6%) of the patients with JSF were treated with a combination of quinolones including levofloxacin (*n* = 9) and ciprofloxacin (*n* = 3). Corticosteroids, G-CSF, immunoglobulin, and thrombomodulin were more frequently administered to patients with SFTS than to those with JSF, reflecting the severity of the disease. Mechanical ventilation was performed in 27.3% of patients with SFTS, which was higher compared to use in patients with JSF (5.3%).

### 3.7. Decision Tree Analysis for Differentiation between JSF and SFTS

To investigate factors that could differentiate between SFTS and JSF at the initial hospital visit, all cases were divided into a 7:3 test cohort (*n* = 44) and a validation cohort (*n* = 17). A decision tree analysis was performed by applying variables with significant differences in bivariate analysis that excluded variables not available in >20% of patients for a training cohort group. In the training cohort, the final classification using decision tree analysis extracted only two variables: WBC < 4000/μL and altered mental status. A WBC < 4000/μL led to SFTS probability of 100% (16/16). In the case that the patients with WBC > 4000/μL have altered mental status, the possibility of SFTS was 14.3% (Figure 1A). For the validation cohort, the area under the curve (AUC) by the Receiver Operating Characteristic (ROC) of the decision tree was 1.000.

In addition, since blood tests are not available quickly in some situations, another decision tree analysis was performed to distinguish SFST from JSF based only on physical findings with significant differences by bivariate analysis among the training cohort and is shown in Figure 2A. A decision tree analysis using only physical findings extracted two variables: skin rash and altered mental status. Absence of skin rash led to SFTS probability of 100% (12/12). When skin rash was present, altered mental status increased the SFTS probability to 50% (2/4). AUC by ROC of the decision tree was 0.871 with 71.4% sensitivity and 90.0% specificity.

Next, we evaluated each variable for which there was a significant difference in bivariate analysis and in cases with missing values less than 20% in Table 2 and Table 3. In the analysis, continuous variables were changed to categorical data, with cutoff values determined for each using the decision tree analysis among the training cohort group described in the first column of Table 5. AUC, sensitivity, specificity, positive predictive value, and negative predictive values were calculated by applying these variables in the validation cohort group (Table 5). In the validation cohort, WBC count < 4000/μL (*p* < 0.0001, AUC 1.000), neutrophil count < 2042/μL (*p* < 0.0001, AUC 1.000), CRP < 4.17 mg/dL (*p* = 0002, AUC 0.833), and platelet count < 64 × 10^3^/μL (*p* = 0.003, AUC 0.818) had excellent discriminatory ability with AUC > 0.8.

Due to the small number of patients in the validation cohort, the decision tree analysis derived from all patients in the study and detected AUC, sensitivity, and specificity of each variable are shown in Appendix A.

## 4. Discussion

SFTS and JSF are tick-borne diseases that are geographically, seasonally, and clinically similar [39]. Definitive diagnosis of these diseases is time-consuming, and only a few university hospitals or specialized facilities in Japan are able to perform SFTSV PCR. Differentiating between these two infections is important because their treatment is different, and more importantly, SFTSV is capable of nosocomial transmission [27,28].

The present study revealed that SFTS and JSF did not differ in patients’ underlying conditions; however, many symptoms, such as altered mental status and diarrhea were more frequently found in patients with SFTS. These symptoms may be associated with viremia in SFTS [30,31]. Skin rash and tick bite were frequently found in patients with JFS. Comparison of chest CT findings between SFTS and JSF revealed that bronchogenic infiltration, such as centrilobular nodules and bronchial wall thickening, was likely to occur more frequently in patients with SFTS, which may be due to secondary bacterial or fungal pulmonary infections [31,40]. Notably, abnormal findings on chest CT were found in around 80% of the SFTS group; ground glass opacity, and interstitial septal thickening was detected in more than 50% of patients. Hospitalized patients with SFTS or JSF frequently develop pulmonary edema that may be caused by systemic inflammation [31,40,41,42]. Our study is the first to compare chest radiologic findings between SFTS and JSF; however, the results indicated that radiographical differentiation between these two infectious diseases was difficult.

Many laboratory data values at admission were different between patients with SFTS and those with JSF. Among them, leukopenia of WBC < 4000/µL, as supported by a previous report [8], was the most remarkable variable that was able to distinguish SFTS from JSF with sensitivity of 100%, specificity of 100%, and AUC 1.000. Peripheral blood count is a basic examination that can be performed in most facilities, and results can be obtained within one hour. Altered mental status was detected in more than 60% of SFTS patients in our study, a slight increase from the previously reported 51% [8]. The possibility of SFTS cannot be ruled out if altered mental status is present even when WBC is above 4000 /μL, based on our cohort. If a patient with suspected tick-borne disease has an elevated WBC (≥4000 /μL) but an altered mental status, it is recommended to test for SFTS at the same time as treating rickettsial disease.

We also attempted to differentiate SFTS from JSF based on physical examination alone, assuming a situation where blood tests are not available immediately. Our results showed that the presence or absence of a skin rash was the most important physical finding in differentiating between the two, and the absence of a skin rash was most likely in SFTS patients (100%). Even in the presence of a rash, altered mental status was associated with a high probability of SFTS (50%).

To our best knowledge, only one previous report discussed the differentiation between JSF and SFTS. Kawaguchi et al. [8] reported that CRP level ≤ 1.0 mg/dL was the most useful variable for differentiating SFTS from JSF. The decision tree in our study did not keep CRP in the final node; however, CRP < 4.17 mg/dL itself had sensitivity of 83.3%, specificity of 90.9%, and AUC 0.833 in our study. In Appendix A, we also performed a decision tree for each variable among all cases in our study, and the cutoff value of CRP 1.78 mg/dL (sensitivity of 87.0%, specificity of 100.0%, and AUC 0.935) for differentiation was similar to that previously reported [8]. Although the incidence of scrub typhus in our study period was low and comparisons could not be made, it is reported that scrub typhus and JSF are both rickettsial diseases and result in similar symptoms and blood test data [43,44]. Table 6 lists previous reports which differentiated between SFTS and other tick-borne infectious diseases such as JSF and scrub typhus. Some previous studies also supported our findings that leukopenia (WBC < 4000 /μL) and altered mental status are significant differentials for SFTS.

Our study had several limitations. First, we enrolled hospitalized patients from secondary and tertiary medical institutions. Among 40 cases of SFTS and 109 cases of JSF diagnosed in our area according to the Infectious Diseases Weekly Report of Nagasaki Prefecture between 2013 and 2020 [45,46], our study included 23/40 (57.5%) of patients with SFTS and 38/109 (34.8%) of patients with JSF. Selection bias, especially for patients with JSF, may have occurred because hospitalized severe cases were mainly included in the study. In fact, our data show a higher percentage with 45.2% of SFTS patients in tertiary medical institutions compared to 21.1% in secondary medical institutions. Second, the data were obtained retrospectively; therefore, missing data values regarding such characteristics as PCT, ferritin, sIL2-R, IgG, and fibrinogen were insufficient for applying to regression analysis and could not be used to determine the significance of these factors in differentiating diseases. In addition, blood tests at the patient’s first visit had variations from the date of infection. This blood time-point of the infection may affect the readout values. Third, scrub typhus, another tick-borne disease, is also prevalent in western Japan, including Nagasaki prefecture, though comparisons could not be made in this study based on the low number of scrub typhus during the study period. In clinical practice, it is necessary to distinguish SFTS from other tick-borne infectious diseases prevalent in the area. A prospective study including mild cases with secured data collection from patients is crucial to substantiate our results and construct a scoring system in real-world practice.

**Table 6 viruses-14-01807-t006:** Comparison of studies to differentiate SFTS from other zoonoses including tick-borne.

Studies	Study Patients	Differentiation Model	Results
Kim MC et al., 2018. [47]	SFTS (*n* = 21) scrub typhus (*n* = 98)	Scoring system (score > 1: SFTS) using 4 factors altered mental status leukopenia (WBC < 4000 /μL) prolonged APTT (>35 s) normal CRP (≤1.0 mg/dL)	score > 1 100% sensitivity 97% specificity AUC 0.995
Sul H et al., 2022. [48]	SFTS (*n* = 183) scrub typhus (*n* = 173)	Scoring system (score > 1: SFTS) using 4 factors leukopenia (WBC < 4000 /μL) prolonged APTT (>40 s), normal CRP (≤3.0 mg/dL) elevated CK level (>1000 IU/L)	score > 1 97% sensitivity 98% specificity AUC 0.992
Kawaguchi T et al., 2020. [8]	SFTS (*n* = 41) JSF (*n* = 40)	CRP ≤ 1.0 mg/dL	95% sensitivity 97% specificity AUC 0.96
Heo DH et al., 2020. [49]	SFTS (*n* = 35) other endemic zoonoses (*n* = 49)	Scoring system (score ≥ 2: SFTS) using 4 factors neurologic symptom diarrhea leukopenia (WBC < 4000 /μL) normal CRP (<0.5 mg/dL)	score ≥ 2 82.9% sensitivity 95.9% specificity AUC 0.950
Our study	SFTS (*n* = 23) JSF (*n* = 38)	Decision tree analysis Pattern A: leukopenia (WBC < 4000 /μL) altered mental status	100% sensitivity 100% specificity AUC 1.000
		Pattern B: absent of skin rash altered mental status	71.4% sensitivity 90.0% specificity AUC 0.871

Abbreviations: SFTS, severe fever with thrombocytopenia syndrome; JSF, Japanese spotted fever; AUC; area under the curve, CI; confidence interval; WBC, white blood cell; APTT, activated partial thromboplastin; CRP, C-reactive protein; CK, creatine kinase.

## 5. Conclusions

In conclusion, when clinicians differentiate between SFTS and JSF among patients with suspected tick-borne infection, a single item, WBC < 4000/µL, strongly differentiates SFTS from JSF. Additionally, the absence of a skin rash and altered mental status may be helpful for differential diagnosis.

## Figures and Tables

**Figure 1 viruses-14-01807-f001:**
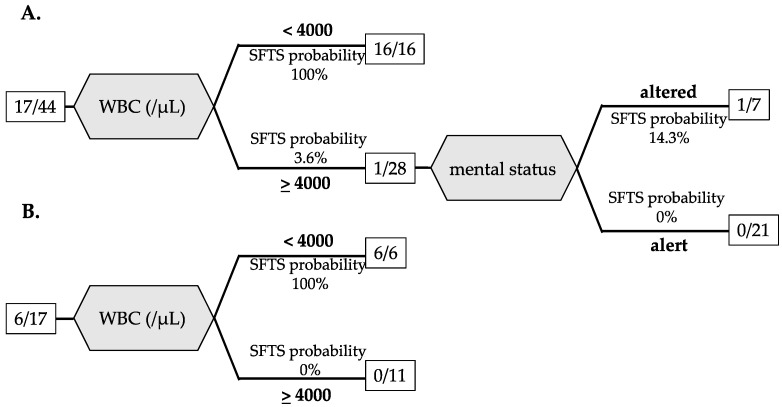
Decision tree classification of SFTS and JSF based on laboratory data and physical examination for the training cohort (**A**) and the validation cohort (**B**). The number in the square is the number of SFTS patients/total patients.

**Figure 2 viruses-14-01807-f002:**
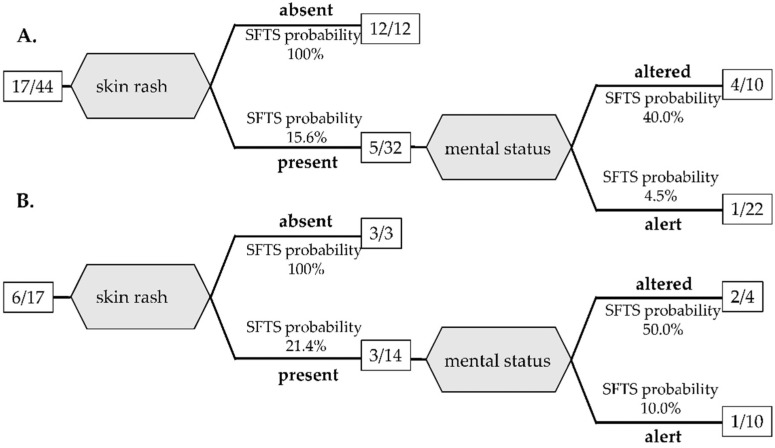
Decision tree classification of SFTS and JSF based on physical examination for the training cohort (**A**) and the validation cohort (**B**). The number in the square is the number of SFTS patients/total patients.

**Table 1 viruses-14-01807-t001:** Clinical characteristics in patients with SFTS or JSF.

	All (*n* = 61)	SFTS (*n* = 23)	JSF (*n* = 38)	*p* Value
Age (years)	69.5 ± 11.0	70.6 ± 10.3	68.8 ± 11.5	0.536
Male Gender	33 (54.1)	15 (65.2)	18 (47.4)	0.197
Farmers, hunters, living or working in wooded and hilly areas	21/58 (36.2)	11/22 (50.0)	10/36 (27.8)	0.101
** *Underlying diseases* **				
Hypertension	30 (49.2)	11 (47.8)	19 (50.0)	1.000
Cardiovascular diseases	9 (14.8)	4 (17.4)	5 (13.2)	0.718
Chronic kidney diseases	6/57 (10.5)	1/21 (4.8)	5/36 (13.9)	0.397
Hemodialysis	1 (1.6)	1 (4.4)	0	0.377
Chronic liver diseases	0 (0.0)	0 (0.0)	0 (0.0)	
Diabetes mellitus	9 (14.8)	5 (21.7)	4 (10.5)	0.278
Solid cancer	7 (11.5)	3 (13.0)	4 (10.5)	1.000
Autoimmune disease	1 (1.6)	1 (4.4)	0 (0.0)	0.377
Immunosuppressant use	2 (3.3)	2 (8.7)	0 (0.0)	0.138
** *Seasonal incidence* **				0.047
Spring (March–May)	20 (32.8)	11 (47.8)	9 (23.7)	
Summer (June–August)	26 (42.6)	10 (43.5)	16 (42.1)	
Autumn (September–November)	15 (24.6)	2 (8.7)	13 (34.2)	

Data are present as number of patients (%) or mean ± SD. Abbreviations: SFTS, severe fever with thrombocytopenia syndrome; JSF, Japanese spotted fever; ICU, intensive care unit. Significant difference level: 0.05/12 = 0.0042.

**Table 2 viruses-14-01807-t002:** Symptoms and physical findings at admission in patients with SFTS or JSF.

	All (*n* = 61)	SFTS (*n* = 23)	JSF (*n* = 38)	*p* Value
Time from onset to admission (days)	5.7 ± 3.5	5.3 ± 4.3	5.9 ± 2.9	0.148
** *General Symptoms* **				
Fatigue	28/60 (46.7)	12/22 (54.6)	16/38 (42.1)	0.425
Headache	13/60 (21.7)	5/22 (22.7)	8/38 (21.1)	1.000
Altered mental status	22 (36.1)	14 (60.9)	8 (21.1)	0.003 †
Myalgia	4/60 (6.7)	1/22 (4.6)	3/38 (7.9)	1.000
** *Respiratory symptoms* **				
Cough	3 (4.9)	1 (4.4)	2 (5.3)	1.000
Sputum	2 (3.3)	1 (4.4)	1 (2.6)	1.000
Dyspnea	11 (18.0)	8 (34.8)	3 (7.9)	0.014
Rales	6 (9.8)	3 (13.0)	3 (7.9)	0.664
** *Gastrointestinal symptoms* **				
Nausea	7/59 (11.9)	5/22 (22.7)	2/37 (5.4)	0.090
Anorexia	29/59 (49.2)	10/22 (45.5)	19/37 (51.4)	0.789
Diarrhea	15/59 (25.4)	11/21 (52.4)	4/38 (10.5)	0.001 †
** *Skin and other symptoms* **				
Skin rash	46 (75.4)	8 (34.8)	38 (100.0)	<0.0001 †
Tick bite	41/57 (71.9)	9/21 (42.9)	32/36 (88.9)	0.001 †
Lymphadenopathy	16/60 (26.7)	10/22 (45.5)	6/38 (15.8)	0.017
** *Vital signs* **				
Body temperature (°C)	38.5 ± 0.9	38.2 ± 0.9	38.6 ± 0.9	0.127
Systolic blood pressure(mmHg)	116.7 ± 21.7	126.1 ± 19.1	110.9 ± 21.3	0.007
Heart rate (/min)	90.8 ± 17.9	84.0 ± 15.7	95.0 ± 18.2	0.020
Respiratory rate (/min) *	20.1 ± 6.54	20.0 ± 6.4	20.2 ± 6.8	0.933
SpO_2_/FiO_2_ ratio	427.9 ± 73.7	414.3 ± 83.3	436.7 ± 66.6	0.888
qSOFA score *	1.1 ± 0.9	1.1 ± 0.6	1.1 ± 1.0	0.936

Data are presented as the number of patients (%) or mean ± SD. Abbreviations: SFTS, severe fever with thrombocytopenia syndrome; JSF, Japanese spotted fever; qSOFA, quick sepsis-related organ failure assessment; † significant difference level: 0.05/15 = 0.0033; * data not available in >20% of patients: SFTS (*n* = 16) and JSF (*n* = 22).

**Table 3 viruses-14-01807-t003:** Laboratory findings on admission in patients with SFTS or JSF.

	Reference Value	All (*n* = 61)	SFTS (*n* = 23)	JSF (*n* = 38)	*p* Value
** *Complete blood count* **					
WBCs (/μL)	3300–8600	5286.6 ± 3514.4	1821.3 ± 1028.8	7383.9 ± 2723.4	<0.0001 †
Neutrophils (/μL)	1830–7250	4265.0 ± 3246.0	1190.8 ± 833.0	6176.0 ± 2655.8	<0.0001 †
Lymphocytes (/μL)	1500–4000	701.7 ± 556.4	460.5 ± 281.3	851.7 ± 631.5	0.002
Atypical lymphocytes (%)	0	0.9 ± 2.3	2.0 ± 3.4	0.2 ± 0.5	0.001 †
Platelets (×10^3^/μL)	158–348	92.4 ± 56.9	58.4 ± 40.1	113.1 ± 56.0	0.001 †
Hb (g/dL)	13.7–16.8	13.7 ± 1.7	14.0 ± 2.0	13.5 ± 1.5	0.245
** *Biochemical examination* **					
CRP (mg/dL)	<0.17	8.0 ± 8.2	0.8 ± 1.2	12.3 ± 7.5	<0.0001 †
PCT (ng/mL) *	<0.05	1.3 ± 2.4	0.3 ± 0.3	2.7 ± 3.3	0.005
sIL-2R (IU/L) *	127–582	2073.7 ± 1754.8	1451.6 ± 631.8	4521.0 ± 2755.7	0.017
Ferritin (ng/mL) *	40–465	6927.7 ± 9422.6	10,205.6 ± 11,311.8	2389.1 ± 1502.4	0.004
IgG (mg/dL) *	870–1700	1055.5 ± 296.3	1181.5 ± 307.2	906.7 ± 209.2	0.020
Alb (g/dL)	4.1–5.1	3.2 ± 0.6	3.3 ± 0.5	3.1 ± 0.7	0.300
AST (IU/L)	13–40	179.2 ± 226.1	328.3 ± 308.2	88.9 ± 65.0	<0.0001 †
ALT (IU/L)	10–42	76.5 ± 64.0	113.5 ± 83.5	54.2 ± 33.7	0.001 †
LDH (IU/L)	124–222	544.8 ± 352.3	801.5 ± 431.9	389.4 ± 156.9	0.001 †
CK (IU/L)	59–248	1266.4 ± 2938.7	2447.4 ± 4331.0	524.0 ± 1081.1	0.004
BUN (mg/dL)	8–20	25.5 ± 19.4	26.1 ± 17.0	25.2 ± 20.9	0.857
Cre (mg/dL)	0.65–1.07	1.1 ± 0.7	1.0 ± 0.6	1.2 ± 0.8	0.459
Acute Kidney Injury		24/60 (40.0)	10/22 (45.5)	14 / 38 (36.8)	0.589
Na (mEq/L)	138–145	132.8 ± 5.5	132.5 ± 6.2	132.9 ± 5.1	0.790
K (mEq/L)	3.6–4.8	3.9 ± 0.5	4.0 ± 0.6	3.9 ± 0.5	0.575
Cl (mEq/L)	101–108	98.8 ± 6.2	99.7 ± 6.5	98.3 ± 6.0	0.370
** *Blood coagulation test* **					
APTT (s)	25.2–34.4	41.2 ± 13.3	48.0 ± 14.9	36.9 ± 10.4	0.001 †
PT-INR	0.85–1.22	1.1 ± 0.1	1.1 ± 0.1	1.1 ± 0.1	0.519
D-dimer (μg/mL)	<1.0	16.1 ± 18.3	14.4 ± 14.7	17.1 ± 20.2	0.601
FDP (μg/mL)	<5	33.8 ± 41.8	28.5 ± 29.2	37.1 ± 48.1	0.563
Fibrinogen * (mg/dL)	168–329	320.8 ± 133.4	212.7 ± 33.4	382.6 ± 129.8	0.001 †
DIC score	0	3.4 ± 2.5	4.1 ± 2.2	3.0 ± 2.5	0.095
** *Urinalysis* **					
Proteinuria	-	45/52 (86.5)	21/22 (95.5)	24/30 (80.0)	0.217
Hematuria	-	41/52 (78.9)	21/22 (95.5)	20/30 (66.7)	0.016

Data are presented as the number of patients (%) or mean ± SD. Abbreviations: SFTS, severe fever with thrombocytopenia syndrome; JSF, Japanese spotted fever; WBC, white blood cells; Hb, hemoglobin; T-Bil, total bilirubin; AST, aspartate aminotransferase; ALT, alanine transaminase; LDH, lactate dehydrogenase; CK, creatine kinase; CRP, C-reactive protein; PCT, procalcitonin; sIL-2R, soluble interleukin-2 receptor; Alb, albumin; IgG immunoglobulin G; BUN, blood urea nitrogen; Cre, creatinine; APTT, activated partial thromboplastin time; PT-INR, prothrombin time test; INR, FDP; fibrin/fibrinogen degradation products; DIC, disseminated intravascular coagulation; † significant difference level: 0.05/30 = 0.0017; * data not available in >20% of patients: PCT, SFTS (*n* = 18) and JSF (*n* = 13); sIL-2R, SFTS (*n* = 14) and JSF (*n* = 4); ferritin, SFTS (*n* = 18) and JSF (*n* = 10); IgG, SFTS (*n* = 13) and JSF (*n* = 11); fibrinogen, SFTS (*n* = 12) and JSF (*n* = 21).

**Table 4 viruses-14-01807-t004:** Chest CT findings in patients with SFTS and JSF.

	All (*n* = 39)	SFTS (*n* = 22)	JSF (*n* = 17)	*p* Value
Time from symptoms onset to chest CT, days ± SD	5.6 ± 3.5	5.8 ± 4.2	5.3 ± 2.3	0.942
Time from admission to chest CT, days ± SD	0.1 ± 0.3	0.2 ± 0.4	0.0 ± 0.0	0.071
Abnormal chest CT findings	32 (82.1)	19 (86.4)	13 (76.5)	0.677
Consolidation	8 (20.5)	5 (22.7)	3 (17.7)	1.000
Ground glass opacity (GGO)	21 (53.9)	12 (54.6)	9 (52.9)	1.000
Interstitial septal thickening	26 (66.7)	15 (68.2)	11 (64.7)	1.000
Centrilobular nodule	10 (25.6)	8 (36.4)	2 (11.8)	0.140
Bronchial wall thickening	9 (23.1)	8 (36.4)	1 (5.9)	0.052
Cardiomegaly	19 (32.8)	7 (31.8)	12 (33.3)	1.000
Pleural effusion	15 (38.5)	5 (22.7)	10 (58.8)	0.045
Pericardial effusion	6 (15.4)	4 (18.2)	2 (11.8)	0.679
Mediastinal lymph node enlargement	3 (7.7)	3 (13.6)	0 (0.0)	0.243
Hepatomegaly	12 (30.8)	6 (27.3)	6 (35.3)	0.730
Splenomegaly	5 (12.8)	3 (13.6)	2 (11.8)	1.000

Data are presented as the number of patients (%) or mean ± SD. Abbreviations: SFTS, severe fever with thrombocytopenia syndrome; JSF, Japanese spotted fever; CT, computed tomography; SFTS, severe fever with thrombocytopenia syndrome; significant difference level: 0.05/14 = 0.0035.

**Table 5 viruses-14-01807-t005:** Sensitivity, specificity, positive predictive value and negative predictive value for differentiating SFTS from JSF.

	Training Cohort (*n* = 44)		Validation Cohort (*n* = 17)							
	SFTS (*n* = 17)	JSF (*n* = 27)	SFTS (*n* = 6)	JSF (*n* = 11)	AUC	Sensitivity	Specificity	PPV	NPV	*p* Value
Altered mental status	10 (58.8)	6 (22.2)	4 (66.7)	2(18.2)	0.742	0.667	0.818	0.667	0.818	0.045
Diarrhea	9/15 (60.0)	2 (7.4)	2 (33.3)	2 (18.2)	0.576	0.333	0.818	0.500	0.692	0.488
Absent of skin rash	12 (70.6)	0 (0.0)	3 (50.0)	0 (0.0)	0.750	0.500	1.000	1.000	0.786	0.006
WBCs < 4000 (/μL)	16 (94.1)	0 (0.0)	6 (100.0)	0 (0.0)	1.000	1.000	1.000	1.000	1.000	<0.0001
Neutrophils < 2042(/μL)	15 (88.2)	0 (0.0)	6 (100.0)	0 (0.0)	1.000	1.000	1.000	1.000	1.000	<0.0001
Atypical lymphocyte ≥ 2%	5 (29.4)	1 (3.9)	3 (50.0)	1 (9.1)	0.750	0.500	1.000	1.000	0.786	0.006
Platelets < 64 (×10^3^/μL)	10 (58.8)	4 (14.8)	6 (100.0)	4 (36.5)	0.818	1.000	0.636	0.600	1.000	0.003
AST ≥ 261 (IU/L)	9 (52.9)	0 (0.0)	4 (66.7)	2 (18.2)	0.742	0.667	0.818	0.667	0.818	0.045
ALT ≥ 97 (IU/L)	8 (47.1)	2 (7.4)	4 (66.7)	3 (27.3)	0.697	0.667	0.727	0.571	0.800	0.113
LDH ≥ 731 (IU/L)	10 (58.8)	0 (0.0)	4 (66.7)	2 (18.2)	0.742	0.667	0.818	0.667	0.818	0.045
CRP < 4.17 (mg/dL)	17 (100.0)	2 (7.4)	5 (83.4)	1 (9.1)	0.871	0.833	0.909	0.833	0.909	0.002
APTT ≥ 51.4 (s)	6 (35.3)	0 (0.0)	2 (33.3)	1/10(10.0)	0.617	0.333	0.900	0.667	0.692	0.254

Abbreviations: SFTS, severe fever with thrombocytopenia syndrome; JSF, Japanese spotted fever; WBC, white blood cell; AST, aspartate aminotransferase; ALT, alanine transaminase; LDH, lactate dehydrogenase; CRP, C-reactive protein; APTT, activated partial thromboplastin; PPV, positive predictive value; NPV, negative predictive value.

## Data Availability

Data are contained within the article.

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
