# Peer review of "Clinical Differentiation of Severe Fever with Thrombocytopenia Syndrome from Japanese Spotted Fever"

_viruses, 2022, doi:10.3390/v14081807_

Round 1

Reviewer 1 Report

Investigators have focused on a very important diagnostic issue between two tick-born illnesses SFTS and JSF. As investigators note, the available data is limited and could be somewhat biased.  I agree that prospective study is to confirm the findings.  I think overall investigators have clearly presented their work and the manuscript is well written.  

I have following comments on this manuscript

Table 1, 2, 3, 4, supp. 1 all should present p-values after multiple testing correction

Line 288: How was altered mental status defined.  Is there a particular test that was performed?

Line 292: List confounding variables

Line 304 I do not agree with this table of analysis.  All the presented values are continuous variables, yet they are analyzed as categorical variables.

Line 353:  Authors should include a limitation that with blood time-point of the infection will affect the readout values.

Line 316: Please revise 1A.  Quality is not very good.

Line 398: Include reference Satoh et al., Journal of Infection and Chemotherapy Volume 23, Issue 1, January 2017, Pages 45-50

Lastly, though the dataset is rather small, it would be interesting to explore for sources of variability in the dataset. I am particularly interested to know how the data groups by the five different prefectures.

Author Response

Reviewer:1

Comments and Suggestions for Authors

Investigators have focused on a very important diagnostic issue between two tick-born illnesses SFTS and JSF. As investigators note, the available data is limited and could be somewhat biased.  I agree that prospective study is to confirm the findings.  I think overall investigators have clearly presented their work and the manuscript is well written.  I have following comments on this manuscript

Reply:

Thank you for reviewing our paper and making insightful comments. We have revised our manuscript to the best of our ability according to your suggestions. Please find our point-by-point responses below.

C1.

Table 1, 2, 3, 4, supp. 1 all should present p-values after multiple testing correction.

R1.

Thank you for your suggestion. As you indicated, we have examined the multiple testing, determined the significance level with the Bonferroni correction, and added it to the footnotes in Tables. Supplementary table 1 was not used for subsequent analysis, therefore, P<0.05 was used as the statistical significance level.

C2.

Line 288: How was altered mental status defined. Is there a particular test that was performed?

R2.

Thank you for your insightful comment. Japan coma scale (JCS) was used to assess altered mental status and JCS >1 was defined as having altered mental status (ref # 33). This definition was added in lines 112-113 in section 2.2. Definitions.

C3.

Line 292: List confounding variables

R3.

Thank you for your insightful suggestion. We have included all variables in the revised manuscript, including lymphocyte, AST, and LDH. The description of the confounding variables have been deleted from the Tables.

C4.

Line 304 I do not agree with this table of analysis.  All the presented values are continuous variables, yet they are analyzed as categorical variables.

R4.

Thank you for your insightful comment. The categorical variables were classified by the cutoff values which was derived from the decision trees performed on the individual continuous variables. Our text was misleading and have been revised in lines 146-147, in section 2.4. Statistical Analysis.

C5.

Line 353:  Authors should include a limitation that with blood time-point of the infection will affect the readout values.

R5.

Thank you for your valuable suggestion. The text was revised as suggested, in lines 392-394, in section 4. Discussion.

C6.

Line 316: Please revise 1A.  Quality is not very good.

R6. Reviewer 2 pointed out the validation cohort, so we have revised the figure separating the test cohort and the validation cohort. The Figures have been simplified with higher quality. Please see Figure 1A.

C7.

Line 398: Include reference Satoh et al., Journal of Infection and Chemotherapy Volume 23, Issue 1, January 2017, Pages 45-50

R7.

The suggested reference has been added as ref #39 in line 332 in the section 4. Discussion.

C8.

Lastly, though the dataset is rather small, it would be interesting to explore for sources of variability in the dataset. I am particularly interested to know how the data groups by the five different prefectures.

R8.

Thank you for your insightful comment. All five hospitals in our data are located in Nagasaki Prefecture, in western part of Japan. There was no difference in the percentage of JSF and SFTS by location within Nagasaki Prefecture. However, the proportion of patients with SFTS was higher at tertiary medical hospitals. These results have been added in lines 385-388.

Reviewer 2 Report

To the authors,

In this paper, Nakada N, et al. conduct a retrospective comparison of clinical features and laboratory data of patients with SFTS and JSF, and they propose a decision tree classification of SFTS and JSF. This is an interesting work to make such a decision tree to differentiate SFTS from JSF, however there are some problems with the decision tree.

Major comments

1.      About the decision tree (Fig.1).

(1)    According to the decision tree in this manuscript, if a patient has an alteration of mental status, the patient's diagnosis will always be “SFTS probable or possible” regardless of the white blood cell count. On the other hand, from previous reports, it is clear that some JSF patients may have central nervous system infection (meningitis) and impaired consciousness. When this decision tree is applied in general medical practice, some JSF patients with impaired consciousness may be misdiagnosed as SFTS. I consider this a serious problem that leads doctors to misdiagnosis. Please state the author's opinion on this risk of misdiagnosis. 

(2)    JSF and SFTS are not the only tick-borne infections in Japan. When this decision tree is used in patients with suspected tick-borne infections, how will patients with scrub typhus, which is as prevalent as JSF, be differentiated? The authors need to elaborate on this point.

(3)    It is presumed that a cross-validation process was performed when creating the decision tree; however, as the authors themselves point out in lines 362-365, the weakness of this manuscript is that the decision tree has not been validated by the validation cohort (patients with suspected tick-borne infections). If the authors want to propose this decision tree, I recommend the authors to validate whether the decision tree is well-fitting or not for practice.

2.      This is a comment. Generally, JSF patients have skin rashes, and SFTS patients do not. In fact, in this study as well, skin rashes were observed in all JFS cases. “Absence of skin rash” seems to be better discriminating factor between SFTS and JSF than “alteration of mental status”. Could the factor of "absence of skin rash" be a useful factor in the decision tree?

Minor comments

1.      Line 56. The latest classification name for SFTS virus is Dabie bandavirus. Please change ‘Huiyangshan bandavirus’ to ‘Dabie bandavirus’

2.      Line 77. The authors should change “mite species” to “tick species”. It is more appropriate.

3.      Table 5 and Table 6. About platelet count. 10^3 should be corrected to 103.

4.      Table 6. Line 312. Please delete the sentence of ‘Data are presented as number of patients (%) or mean +SD.’ It is unnecessary. 

5.      Supplementary Table 1. “(%)” of ICU admission and DIC is unnecessary.

6.   Supplementary Table 1. The abbreviation of CHDF is not correct. The author should describe full spelling of CHDF.  

Author Response

Reviewer 2:

Comments and Suggestions for Authors

In this paper, Nakada N, et al. conduct a retrospective comparison of clinical features and laboratory data of patients with SFTS and JSF, and they propose a decision tree classification of SFTS and JSF. This is an interesting work to make such a decision tree to differentiate SFTS from JSF, however there are some problems with the decision tree.

Reply:

Thank you for reviewing our paper. We appreciate your positive feedback. Please find our responses below, showing our compliance with your valuable suggestions. 

Major comments

C1.

About the decision tree (Fig.1). According to the decision tree in this manuscript, if a patient has an alteration of mental status, the patient's diagnosis will always be “SFTS probable or possible” regardless of the white blood cell count. On the other hand, from previous reports, it is clear that some JSF patients may have central nervous system infection (meningitis) and impaired consciousness. When this decision tree is applied in general medical practice, some JSF patients with impaired consciousness may be misdiagnosed as SFTS. I consider this a serious problem that leads doctors to misdiagnosis. Please state the author's opinion on this risk of misdiagnosis. 

R1.

Thank you for your insightful comment. We agree that some JSF patients have impaired consciousness, as we found 21.1% in our patients. The decision tree analysis in our study found that there was a high probability (100%) of SFTS among patients with WBC <4000 /μL, and even if WBC > 4000 /μL, there was 14.3% probability of SFTS when patient has an altered mental status in the training cohort. We have added these results in lines 28ï¼”-292. Figure 1A was simplified to help readers’ understating.

C2.

JSF and SFTS are not the only tick-borne infections in Japan. When this decision tree is used in patients with suspected tick-borne infections, how will patients with scrub typhus, which is as prevalent as JSF, be differentiated? The authors need to elaborate on this point.

R2.

Thank you for your insightful comment. The prevalence of scrub typhus as well as SFTS and JSF was investigated; however, the occurrence of scrub typhus in Nagasaki prefecture was small during the study period. Based on previous reports, it is assumed that tsutsugamushi disease (scrub typhus) is similar to JSF, not to SFTS. The clinical differentiation of tick-borne infections by previous reports is listed in Table 7 and discussed in lines 374-380, in section 4. Discussion.

C3.

It is presumed that a cross-validation process was performed when creating the decision tree; however, as the authors themselves point out in lines 362-365, the weakness of this manuscript is that the decision tree has not been validated by the validation cohort (patients with suspected tick-borne infections). If the authors want to propose this decision tree, I recommend the authors to validate whether the decision tree is well-fitting or not for practice.

R3.

Thank you for your insightful comment on the methodological ambiguity in the analysis. We agree with your point and we have divided the study population into training cohort (n=44) and validation cohort (n=17) and performed cross-validation process. As you indicated, we have revised the Figures 1 and 2, Supplementary Figures 1 and 2. The Results section has been revised according to the analysis in section 3.7. Decision tree analysis for differentiation between JSF and SFTS.

C4.

This is a comment. Generally, JSF patients have skin rashes, and SFTS patients do not. In fact, in this study as well, skin rashes were observed in all JFS cases. “Absence of skin rash” seems to be better discriminating factor between SFTS and JSF than “alteration of mental status”. Could the factor of "absence of skin rash" be a useful factor in the decision tree?

R4.

Thank you for your valuable comment. Considering the usefulness for clinical practice, we have created another decision tree in Figure 2 based on physical findings only. As suggested, “skin rash” came out as the most differentiable factor between SFTS and JFS. Text has also been added in lines 293-300, in the section 3.7. Decision tree analysis for differentiation between JSF and SFTS.

Minor comments

C5.

Line 56. The latest classification name for SFTS virus is Dabie bandavirus. Please change ‘Huiyangshan bandavirus’ to ‘Dabie bandavirus’

R5.

Thank you for pointing this out. We have revised as suggested.

C6.

Line 77. The authors should change “mite species” to “tick species”. It is more appropriate.

C6.

Thank you for pointing this out. We have revised as suggested.

C7.

Table 5 and Table 6. About platelet count. 10^3 should be corrected to 103.

R7.

Thank you for pointing this out. We have revised as suggested.

C8.

Table 6. Line 312. Please delete the sentence of ‘Data are presented as number of patients (%) or mean +SD.’ It is unnecessary. Supplementary Table 1. “(%)” of ICU admission and DIC is unnecessary.

R8.

Thank you for pointing this out. We have revised as suggested.

C9.

Supplementary Table 1. The abbreviation of CHDF is not correct. The author should describe full spelling of CHDF.  

R9.

Thank you for pointing this out. We have revised as suggested.

Reviewer 3 Report

I read with interest the manuscript authored by Nana Nakada at al. In my opinion the paper is worth publishing. It describes in a chronological manner clinical differences between STFS and JFS based on many important factors such as patients’ baseline characteristics, clinical symptoms, laboratory findings and radiological findings. The discussion needs major revisions. The authors have to discuss obtained results with those published by other researchers. In its current form discussion repeats results. I strongly recommend to rewrite this section.

Some minor correction which are required :

L 77 – Haemaphysalis not Haemaohysalis

L 185 – JSF not JFS (the same in L191, 216, 230, 231)

Table 4 – centrilobular nodule %  - 10 from 36 is 27.7% not 25%

the same with pleural effusion % - 14 from 36 is 38.8% not 22.7%

Author Response

Reviewer3:

Comments and Suggestions for Authors

I read with interest the manuscript authored by Nana Nakada at al. In my opinion the paper is worth publishing. It describes in a chronological manner clinical differences between STFS and JFS based on many important factors such as patients’ baseline characteristics, clinical symptoms, laboratory findings and radiological findings.

Reply:

Thank you for reviewing our paper and making insightful comments. We have revised our manuscript to the best of our ability according to your suggestions. Please find our point-by-point responses below.

C1.

The discussion needs major revisions. The authors have to discuss obtained results with those published by other researchers. In its current form discussion repeats results. I strongly recommend to rewrite this section.

R1.

Thank you for the important suggestion. Table 6 was added to compare the results with those presented by other researchers regarding differentiation of tick-borne diseases. Figure 2 was added to demonstrate the decision trees using only the physical findings. The text of the discussion section was also revised based on these findings and limitations.

C2.

Some minor correction which are required : L 77 – Haemaphysalis not Haemaohysalis

R2.

Thank you for pointing this out. We have revised as suggested.

C3.

L 185 – JSF not JFS (the same in L191, 216, 230, 231)

R3.

Thank you for pointing this out. The errors have been corrected.

C4.

Table 4 – centrilobular nodule %  - 10 from 36 is 27.7% not 25%

R4.

Thank you for pointing this out. The errors have been corrected.

C5.

the same with pleural effusion % - 14 from 36 is 38.8% not 22.7%

R5.

Thank you for pointing this out. The errors have been corrected.

Round 2

Reviewer 2 Report

In this revised paper, Nakata et al. improve the decision tree to distinguish between SFTS and JSF. I would like to request the author to reconsider the decision tree.

Comments

1.        About the decision trees. The authors divide the dataset into two cohorts, a training cohort and a validation cohort, and analyze the decision trees. I agree that “WBC <4000 μl” would be a good variable for splitting the root node. On the other hand, altered mental status was found in 14 of 23 SFTS patients and 8 of 38 JSF patients in the dataset, suggesting that altered mental status is a rather common symptom of both diseases. Is it really a good idea to use “mental status” as a variable for splitting in the decision tree? Since blood count tests are readily available at most medical centers, can't you use "skin rash" (you stated that “skin rash” is as the most differentiable factor between SFTS and JFS) and "WBC <4000 μl" as variables to create a better decision tree? Please consider it again; however, if the result of the reconsideration is not appropriate scientifically, you can dismiss my suggestionts are readily available at most medical centers, can't you use "skin rash" (you stated that “skin rash” is as the most differentiable factor between SFTS and JFS) and "WBC <4000 μl" as variables to create a better decision tree? Please consider it again; however, if the result of the reconsideration is not appropriate scientifically, you can dismiss my suggestion

2.        Line 377. Table 7 is incorrect. (Table 6?)

Author Response

Reviewer2:

In this revised paper, Nakata et al. improve the decision tree to distinguish between SFTS and JSF. I would like to request the author to reconsider the decision tree.

 Thank you for reviewing our paper and making insightful comments. We have revised our manuscript based on your suggestions. Please find our point-by-point responses below.

Comments

C1.

About the decision trees. The authors divide the dataset into two cohorts, a training cohort and a validation cohort, and analyze the decision trees. I agree that “WBC <4000 μl” would be a good variable for splitting the root node. On the other hand, altered mental status was found in 14 of 23 SFTS patients and 8 of 38 JSF patients in the dataset, suggesting that altered mental status is a rather common symptom of both diseases. Is it really a good idea to use “mental status” as a variable for splitting in the decision tree? Since blood count tests are readily available at most medical centers, can't you use "skin rash" (you stated that “skin rash” is as the most differentiable factor between SFTS and JFS) and "WBC <4000 μl" as variables to create a better decision tree? Please consider it again; however, if the result of the reconsideration is not appropriate scientifically, you can dismiss my suggestions.

R1.

Thank you for valuable comments. We understand your considerations. However, in the present study, the decision tree analysis resulted in selecting two factors, WBC < 4000/μL and altered mental status, as a final node. This may due by the only one patient with SFTS having WBC > 4000 /μL in the training cohort showed both altered mental status and skin rash. Although absent skin rash was the most important differential physical finding of SFTS as shown in Figure 2, it did not remain in the final node by automatic statistical analysis (scientifically fare analysis) in our study in Figure 1. A future study with larger number of cases may change a decision tree as you proposed.

 C2.   Line 377. Table 7 is incorrect. (Table 6?)

R2.

Thank you for pointing this error out. We have corrected Table 7 to Table 6 as you indicated.

Reviewer 3 Report

The Authors revised the manuscript and it may be published in present form.

The Figures seem to be much more clear in the present layout design – please only standardize the description of  the Figures - L 315 and L 319  according to the Publisher's guidelines (bold or without bold).

Author Response

Reviwer3:
The Authors revised the manuscript and it may be published in present form.

Thank you for reviewing our paper. We appreciate your positive feedback.

C1.

The Figures seem to be much more clear in the present layout design – please only standardize the description of the Figures - L 315 and L 319 according to the Publisher's guidelines (bold or without bold).

R1.

Thank you for pointing this out. We have revised the text as suggested.